# Characterization and Actuation of Ionic Polymer Metal Composites with Various Thicknesses and Lengths

**DOI:** 10.3390/polym11010091

**Published:** 2019-01-08

**Authors:** Shufeng Li, Joanne Yip

**Affiliations:** 1Key Laboratory of Advanced Textile Composites, College of Textile Science and Engineering, Tianjin Polytechnic University, Tianjin 300387, China; 2Institute of Textiles and Clothing, The Hong Kong Polytechnic University, Hung Hom, Kowloon 00852, Hong Kong; joanne.yip@polyu.edu.hk

**Keywords:** IPMC, actuation, thickness, length, repeated actuations

## Abstract

Ionic polymer metal composites (IPMCs) with various thicknesses of 1, 2, 4, and 6 nafion films (denoted as 1-film, 2-film, 4-film and 6-film, respectively) are fabricated, and their characterization and actuation performances are then investigated. The effects of the thickness of the IPMCs on their morphology, surface resistance, and water uptake capability are studied. Their actuation performances are further evaluated by examining the tip force and displacement in terms of the length and the thickness of the IPMCs, under a direct current (DC) power of 3.0 or 4.5 V. In comparison with the 1-film, the 2-film shows a six-fold increase in the maximum tip force, but the response time increases from 2 to 9 s. The 4-film doubles the maximum tip force of the 2-film at 21 s. On the other hand, a reduction of the length of the IPMC from 30 to 15 mm also results in a double-maximum tip force, but this never increases the response time. Repeated actuations of the IPMCs with various thicknesses are performed by three actuation methods of no treatment, treatment in deionized water, and treatment in a NaCl solution. The relationships between the repeated actuation methods and actuations of the IPMCs with various thicknesses are also investigated.

## 1. Introduction

Ionic polymer metal composites (IPMCs) are one of the most promising smart materials, due to their larger displacements under lower voltages, and they can be potentially applied as biomimetic sensors, robotic actuators, or artificial muscles [1,2]. The IPMC actuators mainly consist of a perfluorinated ion exchange membrane that is sandwiched between two metal electrodes. When the input voltage is applied, the IPMC bends toward the anode. The bending behavior generates the actuation force and displacement. However, the low actuation forces of the IPMCs have limited their applications in actuation.

Many efforts have been made to improve the actuation forces of the IPMCs by adding nano-metal powders [3], graphene [4,5], carbon tubes [6,7], and carbon nanofibers [8] to enhance the performance of the ion exchange polymers, synthesizing different materials to produce new types of ion exchange membranes [9,10,11,12,13,14,15], using ionic liquids to improve the durability of the IPMCs [16,17], conducting electroplating processes to deposit different materials for use as surface electrodes with exceptional performance [18], carrying out plasma pretreatment on the surface of nafion to improve the adhesion between the nafion film and the Pt electrode layer [19,20], coating a thin film of parylene to suppress the water evaporation from the IPMC [21], and alternating with electroless plating [22]. However, all of these methods require careful fabrication, or they involve complicated chemical treatments.

Recently, Dr. Ma [23] enhanced the tip force of the IPMCs by manufacturing the thicker IPMCs, through casting of the nafion solution with strictly-controlled casting processes. Furthermore, Dr. Lee [24] proposed a new fabrication method for the IPMC actuators, by hot-pressing a number of nafion films to fabricate the thick IPMCs. They had to carry out multiple cycles of electroless plating with platinum (Pt), to obtain a better actuation performance. However, how the thick IPMCs influence the response time to the tip force and displacement was not discussed.

In this paper, IPMCs with various thicknesses are fabricated by the hot-pressing method, and their morphologies, surface resistance, water uptake capabilities, and actuation performances are examined. The effects of IPMC thickness on actuation are systematically investigated. The relationships among the thickness, maximum tip force, or displacement and the response time are determined, and an appropriate thickness is proposed. In addition, effects of the lengths of the IPMCs on actuation are taken into account. Finally, repeated actuations of the IPMCs with various thicknesses are discussed.

## 2. Materials and Methods

### 2.1. Materials

*A* nafion^®^-117 film with a thickness of 0.18 mm was obtained from Dupont Co., Shanghai Branch, China, and used as the ionic polymer membrane. Tetraammineplatinum (II) chloride hydrate ([Pt(NH_3_)_4_]Cl_2_, 98.0%) was purchased from Sigma-Aldrich Co., Shanghai Branch, China. Sodium borohydride, hydrazine monohydrate, hydroxylamine hydrochloride, hydrochloric acid, and sodium chloride were obtained from Tianjin Chemical Reagent Co., Ltd., Tianjin, China. All of the materials were used as received without further purification.

### 2.2. Fabrication of the IPMCs

Four IPMC specimens with various thicknesses were fabricated by hot-pressing 1, 2, 4, and 6 nafion films [24], which are referred to as 1-film, 2-film, 4-film, and 6-film, respectively. The fabrication of the IPMCs was carried out as follows.

(1) The nafion films were cut into dimensions of 5 × 5 cm^2^, and stacked together. Then, they were placed into a hot-press mold at 180 °C, held without pressure for 10 min, and then pressed at 180 °C under a pressure of 7 bars for 10 min.

(2) The surfaces of the pressed films were roughened with fine sand paper. Then, they were placed into an ultrasonic water tank for 1 h, and boiled for 30 min, respectively, in 2 M hydrogen chloride (HCl) and deionized water.

(3) The electroless plating process was performed in accordance with the method in our previous work [22], and is described below.

a. The first reduction. The nafion films were immersed overnight into an aqueous platinum ammine complex solution of [Pt(NH_3_)_4_]Cl_2_. They were placed into 100 mL of stirring water at 40 °C, and 5 mL of a 5 wt % sodium borohydride solution (NaBH_4_) was added every 10 min for a time period of 4 h. Then, the temperature was gradually increased to 60 °C, and an additional 20 mL of 5 wt % NaBH_4_ was added into the mixture, and the solution was stirred for 30 min. Finally, the nafion films were immersed into a 0.1 M HCl solution for 1 h, and then rinsed with deionized water.

b. The second reduction. The specimens were immersed into an aqueous solution of [Pt(NH_3_)_4_]Cl_2_ overnight, and placed into 100 mL of stirring water at 40 °C, and 3 mL of 20 wt % hydrazine monohydrate (NH_2_NH_2_·1.5H_2_O) and 6 mL of 5% hydroxylamine hydrochloride (NH_2_OH·HCl) were added every 30 min until the reaction ended [22]. Finally, the nafion films were immersed into 0.1 M HCl for 1 h, and rinsed with deionized water.

(4) The samples were immersed into a 2 M NaCl solution for preservation.

### 2.3. Scanning Electron Microscopy

A Philips XL 30 environmental scanning electron microscope was used to observe the morphologies of the IPMCs. Before taking the scanning electron microscopy (SEM) micrographs, the IPMC specimens were dried in a vacuum oven at 70 °C for 24 h, to completely remove the water. Energy-dispersive X-ray (EDX) data was measured by using an Inca Energy 300 (Oxford Instrument Co., Ltd., Abingdon, UK) ray instrument on an area of 100 μm^2^ from the surface of the IPMCs, to provide the Pt content.

### 2.4. Surface Resistance

The surface resistance for each IPMC sample was tested between two points at a distance of 1 cm under dry conditions, and the average value over 10 measurements was adopted.

### 2.5. Water Uptake Capability

The IPMC samples were soaked into deionized water at room temperature for 24 h, to absorb a maximum amount of water. The water uptake capability was gravimetrically determined with the ratio of the mass increment in the wet to the mass in the dry.

### 2.6. Actuation Performance

An IPMC actuator was cut into dimensions of 6 mm in width and 45 mm in length, to determine the actuation performance, unless otherwise stated. The tip force and the displacement of the IPMCs were tested in an open atmosphere, and the average value over three measurements was adopted. One edge of each IPMC was clamped, and the tip force was measured by using a load cell system with an accuracy of 0.0001 g [25]. A digital camera was used to record the displacement of the IPMCs under an applied voltage, according to Figure 1.

### 2.7. Repeated Actuations of the IPMCs

The IPMCs with various thicknesses were actuated at a direct current (DC) power of 4.5 V for 1 min, and the actuation was repeated 10 times. Before each actuation, the IPMCs were respectively treated with none, deionized water, and a 2 M NaCl solution for 5 min. The maximum tip force and the maximum displacement were recorded, according to the procedure described in Section 2.6.

## 3. Results and Discussion

### 3.1. Morphology

The morphologies of the IPMCs after electroless plating were observed by SEM. Figure 2 reveals the surfaces of the IPMCs with various thicknesses, and their EDX values. After electroless plating, a Pt metal electrode layer was observed on each IPMC and the presence of Pt on the surface of the IPMCs was confirmed by EDX measurements. The 1-, 2-, and 4-films had similar Pt contents of about 85%, and the 6-film had the lowest Pt content of 79%. This might be due to the fact that the 6-film becomes more compact and stiffer after hot–pressing, which leads to fewer Pt cations being absorbed. The Pt content influences the conductivity of the IPMCs. That is, a greater Pt content results in a better conductivity.

On the other hand, the surfaces of the four IPMCs showed different morphologies. The 1-film has a relatively smooth surface, with some tiny cracks. The 2-film reveals more protrusions, and wider and deeper cracks. Compared with the 2-film, the 4-film has some folds, and fewer cracks on the surface, and the spaces among the folds are broad and shallow. For the 6-film, more folds are densely packed, the space among the folds are narrower and deeper, and cracks disappear. These can be explained by the incompatibility between the Pt particles and the nafion films, caused by the increasing thickness of the IPMCs. In our experiments, it was found that hot-pressing resulted in increased stiffness of the nafion films, which increased with increasing thicknesses. Furthermore, it was also noticed that the morphologies of the IPMCs were observed by SEM at room temperature after they were dried at 70 °C. The decrease in temperature led to the shrinkage of the IPMCs with various thicknesses to different degrees, and dehydration led to the crack-forming on the Pt surface. The thicker and stiffer IPMCs showed more shrinkage. For the 1- and 2-films, they shrank less, so that some cracks and protrusions were readily observed. When the thickness was greater than two nafion films, the IPMCs showed immense shrinkage; the sizes of the cracks reduced, and eventually, the cracks were eliminated, and folds appeared on the surface.

An even and smooth Pt surface suggests good adhesion between the nafion film and the Pt particles, which promotes current passage through the IPMCs under the applied voltage, and improves the actuation performance.

### 3.2. Water Uptake Capability

The IPMCs actuate mainly due to the oriented migration of the hydrated cations in the aqueous solution under an applied voltage. The water in the IPMCs contributes to the migration of the hydrated cations, and reduces the migration barrier. Figure 3 reveals the relationship between the water uptake capability and the thickness of the IPMCs. After hot-pressing, the thickness of the IPMCs became slightly less than that of the stacked nafion films. In addition, it was observed that an increasing thickness led to a reduction in water uptake capability. Compared with a water uptake capability of 25% for the 1-film, the 2- and 4-films had water uptake capabilities of 23.9 and 23.6%, respectively, and these were slightly higher than 23.3% for the 6-film. This is attributed to a more compact structure in the 6-film, caused by hot-pressing. On the other hand, a greater thickness of the IPMCs also extends the migration distance of the hydrated cation transfer, as well as the migration barrier. Therefore, it is supposed that thicker IPMCs with a low water uptake capability will have a slow actuation.

### 3.3. Surface Resistance

The IPMCs are fabricated by depositing a Pt metal electrode layer onto the surface of the nafion films through electroless plating. In the present paper, the electroless plating process is performed by sequential chemical reductions with NaBH_4_ and NH_2_–NH_2_. The relationships between the average surface resistances of the Pt particles and the thicknesses of the IPMCs are shown in Figure 4. It is observed that after the first chemical reduction by NaBH_4_, the 1- and 2-films had an average surface resistance of about 3 Ω, and the 4- and 6-films were about 5 Ω. After the second reduction by NH_2_–NH_2_, the average surface resistances of the four IPMCs were reduced. The 1-film showed the greatest decrease, by 50% to 2 Ω, and the 2- and 4-films, respectively decreased to 2.3 and 3.36 Ω. The 6-film decreased the least for average surface resistance, by 16% to 4.25 Ω, indicating the worst conductivity. It is concluded that an increasing thickness of the IPMCs leads to a larger surface resistance, which corresponds with the results in Section 3.1 that the thicker IPMCs have a more uneven surface, and a lower Pt content.

### 3.4. Effect of IPMC Thickness on Actuation

The thickness of the IPMC impacts the actuation performance. As is well known, an increasing thickness increases the amount of freely-moving cations, and potentially increases the tip force. On the other hand, it extends the distance over which the hydrated cations transfer, enlarges the migration barrier, and leads to a delayed response to the maximum tip force. In this section, how the thickness of the IPMCs influences the tip force, displacement, and their response time is investigated in detail, and the appropriate thicknesses of the IPMCs are discussed.

#### 3.4.1. Effect of IPMC Thickness on Tip Force

Figure 5 reveals the plotted tip force-time curves of the IPMCs with various thicknesses under different DC voltages. At 3.0 V (Figure 5a), the tip forces of the 1-, 2-, 4-, and 6-films all peaked and then decreased. The 1-film peaked at the tip force of 1.37 mN within 2 s. The 2-film showed a maximum tip force of 10.79 mN, but it required 9 s to reach it. The 4-film required an even longer time of 21 s to reach the maximum tip force of 28.24 mN, which was more than twice that of the 2-film. Distinctively, the tip force of the 6-film slowly and steadily increased to the maximum tip force of 22.62 mN at 62 s. 

A similar experimental result was observed at 4.5 V (Figure 5b). The 1-, 2-, 4-, and 6-films required less time to reach the maximum tip force, and then the tip forces decreased. The 6-film showed the greatest maximum tip force of 47.90 mN at about 60 s.

Figure 5c reveals the relationship between the maximum tip force, or the response time to the maximum tip force and the thickness of the IPMCs under different voltages. At a voltage of 3.0 V, the maximum tip force and the thickness showed an appropriate linear relationship until the thickness increased to four films, and then the maximum tip force decreased. A larger voltage of 4.0 V resulted in an increase in the maximum tip force for the four IPMCs, and the 6-film showed the greatest increase. It was also observed that the response time to the maximum tip force increased as the thickness increased, at either 3.0 or 4.5 V. Therefore, to effectively adjust the maximum tip force by thickness, two or four films are recommended, and the film number should be no more than four. Otherwise, more time has to be required to reach the maximum tip force.

#### 3.4.2. Effect of IPMC Thickness on Displacement

Figure 6 shows the displacement–time relationship for the IPMCs with various thicknesses under different voltages. The thicker IPMCs showed slower and less displacements, and the thinner IPMCs had quicker and greater displacements. At 3.0 V (Figure 6a), the 1- and 2-films required, respectively, 2 and 5 s to reach the maximum displacement of about 27 mm. They then behaved differently. The displacement of the 1-film peaked and quickly decreased, sometimes even turning over the zero and then slightly increasing. However, the 2-film peaks in the displacement, and then the displacement slowly decreased. The 4- and 6-films continuously increased to the maximum displacement of 21.92 mm at 50 s, and 7.15 mm at 55 s, respectively.

Figure 6b reveals that a larger voltage of 4.5 V dramatically improved the displacements of the four IPMCs. The 1-film peaked in displacement, and then the displacement slightly decreased and then continued to increase to the maximum value of 120 mm. The displacement of the 4-film exceeded the 2-film at about 20 s, and then it continuously increased to the maximum value of 65 mm at 50 s. The 6-film increased slowly to reach the maximum displacement of 37 mm at 55 s. Meanwhile, these experimental results manifested that a large voltage contributed to a decreasing displacement deviation.

Figure 6c shows the relationships between the maximum displacement or the response time to the maximum displacement and the thickness of the IPMCs under different voltages. It was noticed that at 3.0 V, the 1-, 2-, and 4-films had similar maximum displacements, and the 6-film showed a lower maximum displacement. Regardless of whether the voltage was 3.0 or 4.5 V, the 1- and 2-films reached maximum displacements within 30 s, whereas the 4- and 6-films reached a maximum displacement at about 50 s. Thus, to obtain a larger displacement within 30 s, the thickness should be no more than two films.

### 3.5. Effect of IPMC Length on Actuation

In this section, effects of the lengths of the IPMCs on actuation performances are investigated. The 2-film with a width of 6 mm is used as an example. The IPMCs were cut to lengths of 15, 30, and 45 mm. 

#### 3.5.1. Effect of IPMC Length on Tip Force

Figure 7a illustrates the tip force–time relationships of the IPMCs with lengths of 15, 30, and 45 mm at 3.0 V. It was observed that the tip forces of the three IPMCs all rapidly reached the maximum value at 7 s, and then decreased. The IPMCs with lengths of 30 and 45 mm had maximum tip forces of 10.76 and 10.25 mN, respectively. In comparison, the IPMC with a length of 15 mm showed a double-maximum tip force of 27.50 mN. 

A similar experimental result was observed at 4.5 V in Figure 7b. Three IPMCs with lengths of 15, 30, and 45 mm respectively, reach the maximum tip forces of 37.56, 16.44, and 17.26 mN at about 4 s. Figure 7c demonstrates the relationship between the maximum tip force, or the response time to the maximum tip force, and the length of the IPMCs. The results designated that 30 mm is a crucial length for the IPMCs. When the length exceeds 30 mm, the maximum tip force just slightly increases. However, a reduction of the length from 30 to 15 mm resulted in a double-maximum tip force. More importantly, the reduction of the lengths of the IPMCs never distinctly extended the response time to the maximum tip force, which is particularly significant, because enhancement of the maximum tip force of the IPMCs by thickness is usually at the expense of the response time.

#### 3.5.2. Effect of IPMC Length on Bending Angle

Considering that the lengths of the IPMCs influence the bending distance, the bending angle in degrees, in place of the bending distance in millimeters, is used in this section to evaluate the actuations for the IPMCs with various lengths.

Figure 8 shows the bending angle–time curves of the IPMCs, with various lengths under different voltages. Figure 8a reveals that at 3.0 V, the shorter IPMC showed a lower maximum bending angle. The IPMCs with lengths of 30 and 45 mm required 5 s to reach their maximum bending angles of 17.8° and 25°, respectively. The IPMC with a length of 15 mm required 10 s to achieve a maximum bending angle of 11.2°. Then, they all decreased in bending angle.

Figure 8b shows that at 4.5 V, three IPMCs with various lengths all reached a larger maximum bending angle. The IPMCs with lengths of 30 and 45 mm, respectively, required 10 and 25 s, to reach a maximum bending angle of about 49°, and the IPMC with a length of 15 mm demonstrated a maximum bending angle of 45° at 10 s.

Figure 8c shows the relationship between the maximum bending angle or the response time to the maximum bending angle and the lengths of the IPMCs. It was found that the maximum bending angle increased as the length of the IPMCs increased. However, the response time to the maximum bending angle depends on the applied voltage.

### 3.6. Repeated Actuations

In this section, the IPMCs with various thicknesses were repeatedly actuated. Before each actuation, three actuation methods with no treatment, treatment in deionized water, and treatment in a 2M NaCl solution for 5 min, are respectively used to investigate the repeated actuation of the IPMCs with various thicknesses.

#### 3.6.1. Effects of Repeated Actuation Methods on the Maximum Tip Force

Figure 9 illustrates the maximum tip forces when the IPMCs with various thicknesses were repeatedly actuated 10 times. For the 1-film, three actuation methods of no treatment, treatment in deionized water, and NaCl solution showed similar maximum tip forces. The 2-film evidently manifested a lower maximum tip force loss in treatment with NaCl solution than no treatment, and the treatment in deionized water. When the 2-film was repeatedly actuated after seven times, the maximum tip force with the treatment in NaCl solution still approximately retained 70%, and that without treatment decreased to 40%. However, treatment in deionized water resulted in a greater maximum tip force loss than no treatment. When the 4-film was repeatedly actuated, treatment in a NaCl solution showed the least maximum tip force loss. Meanwhile, it was observed that the maximum tip force loss with the treatment in deionized water was close to that with no treatment. For the 6-film, treatments in NaCl solution and deionized water reveal a similar maximum tip force, and their maximum tip forces were much greater than those with no treatment. The difference between the treatment in NaCl solution and treatment in deionized water is that after the 6-film was repeatedly actuated eight times, the maximum tip force of the former slowly decreased from 60 to 46%, and the latter quickly reduced from 60 to 2%.

Our experimental results indicate that water and hydrated cations contribute to maintaining the tip force when the IPMCs are repeatedly actuated, which is related to the thickness of the IPMCs. When the thickness of the IPMCs increases, the hydrated cations play an important role in retaining the tip force, and the water has negligible effects. When the thickness of the IPMCs is over 4 films, both the water and the hydrated cations effectively decrease the loss of the maximum tip force, resulting in a similar maximum tip force.

#### 3.6.2. Effects of Repeated Actuation Methods on the Maximum Displacement

Figure 10 depicts the maximum displacements of the IPMCs with various thicknesses when they are repeatedly actuated 10 times. Figure 10a shows that the 1-film without treatment had the least maximum displacement loss, followed by the treatment in deionized water, and finally the treatment in NaCl solution. When the 1-film was actuated 10 times without treatment, its maximum displacement still maintained 80%. When the 1-film was actuated five times, the maximum displacement with the treatment in deionized water was still nearly double the treatment in NaCl solution. After that, the maximum displacement with the treatment in deionized water decreased more quickly, and it approximated the value with the treatment in NaCl solution after being actuated seven times. Finally, both of the maximum displacements with the treatments in deionized water and NaCl solution decreased to about 5%.

When the 2-film was repeatedly actuated, treatment in the NaCl solution showed a greater maximum displacement than no treatment and treatment in deionized water. Its maximum displacement still maintained 40%, even though it was repeatedly actuated 10 times. For the 4-film, the three actuation methods showed similar maximum displacement, and the treatment inNaCl solution retained a slightly greater maximum displacement than no treatment and treatment in deionized water. However, the 6-film behaved differently. Treatment in deionized water distinctly displayed a greater maximum displacement than no treatment or treatment in NaCl solution.

It is concluded that no treatment is the best way for the 1-film to maintain the maximum displacement when it is repeatedly actuated. When the thickness of the IPMCs increases, the hydrated cations should be supplemented, to obtain the greater maximum displacement. When the thickness exceeds four films, the water, in place of the hydrated cations, has to be used to achieve a greater maximum displacement.

In summary, when the IPMCs are repeatedly actuated, both water and hydrated cations are beneficial for achieving the greater maximum tip force, and maximum displacement. The hydrated cations are more effective when the thickness is no more than four films, and after that, the water has to be added, because the hydrated cations only contribute to a greater maximum tip force.

## 4. Conclusions

In this study, the IPMCs with various thicknesses and lengths have been fabricated, and their properties, such as morphology, resistance, water uptake capability, and actuation are studied. The thicker IPMCs reveal the deeper cracks, larger folds, greater surface resistances, and less water uptake capabilities. The thicknesses and lengths of the IPMCs impact on the actuation performances. Increasing the thickness greatly enhances the maximum tip force, but it also prolongs the response time. To obtain both a great maximum tip force and a quick response time, the thickness of the IPMCs is recommended to be no more than four films. A reduction of the IPMC length of 30 to 15 mm achieves a double-maximum tip force, but it never extends the response time, which is especially valuable in cases that require a quick response. Actuation methods also influence the repeated actuation of the IPMCs with various thicknesses. To maintain both the greater maximum tip force and the maximum displacement, hydrated cations are more effective for a thickness of no more than four films. When the thickness is over four films, water, rather than the hydrated cations, should be used.

## Figures and Tables

**Figure 1 polymers-11-00091-f001:**
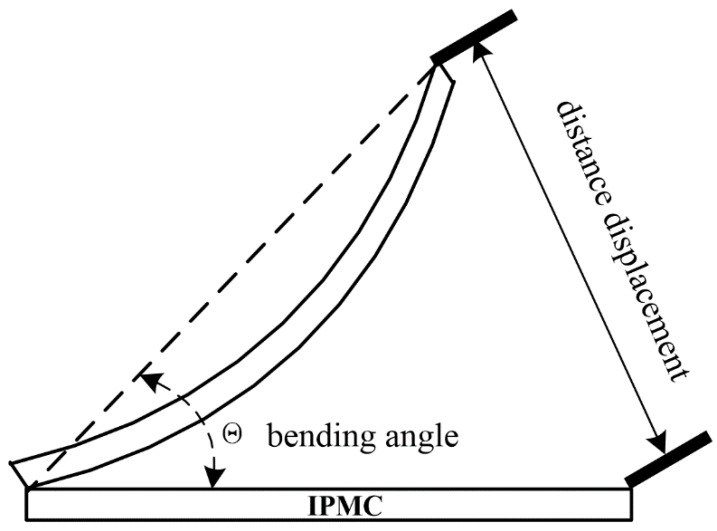
Schematic diagram of the displacement measurement system of the IPMCs.

**Figure 2 polymers-11-00091-f002:**
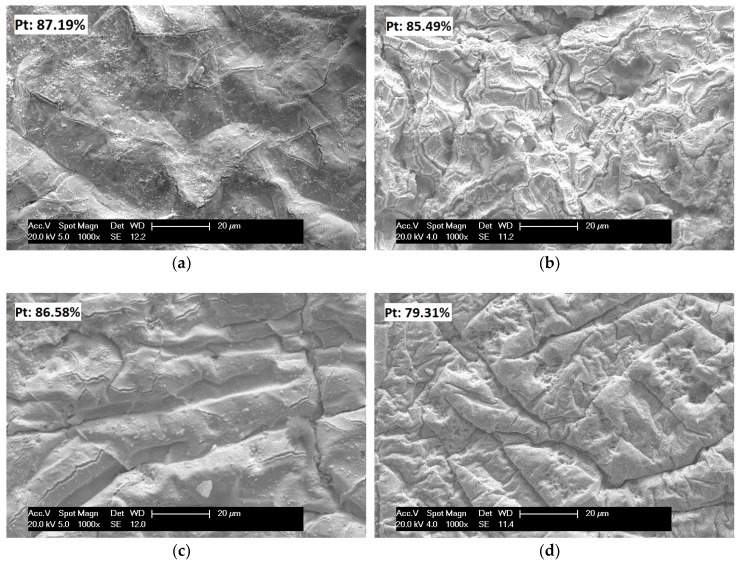
Surface morphology of the IPMCs with various thicknesses. (**a**) 1-film; (**b**) 2-film; (**c**) 4-film; (**d**) 6-film.

**Figure 3 polymers-11-00091-f003:**
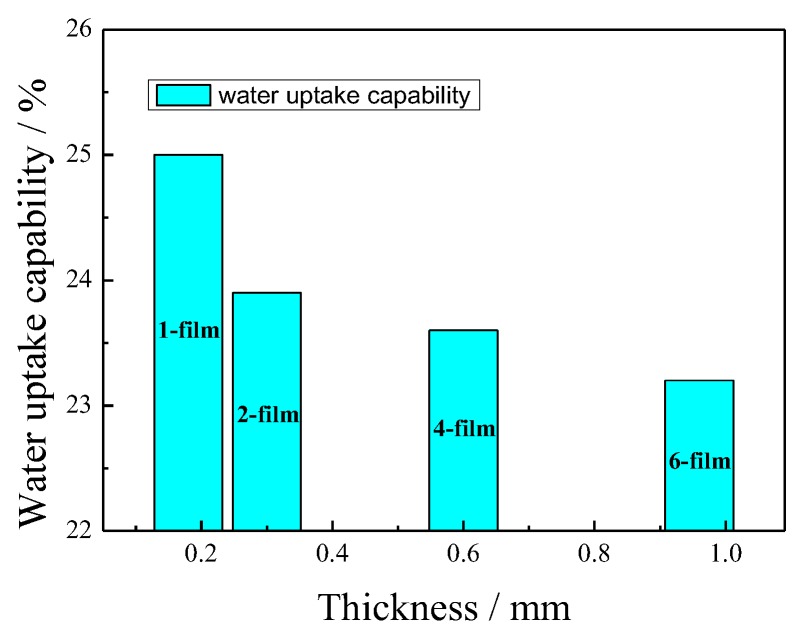
The relationship between the water uptake capability and the thickness of the IPMCs.

**Figure 4 polymers-11-00091-f004:**
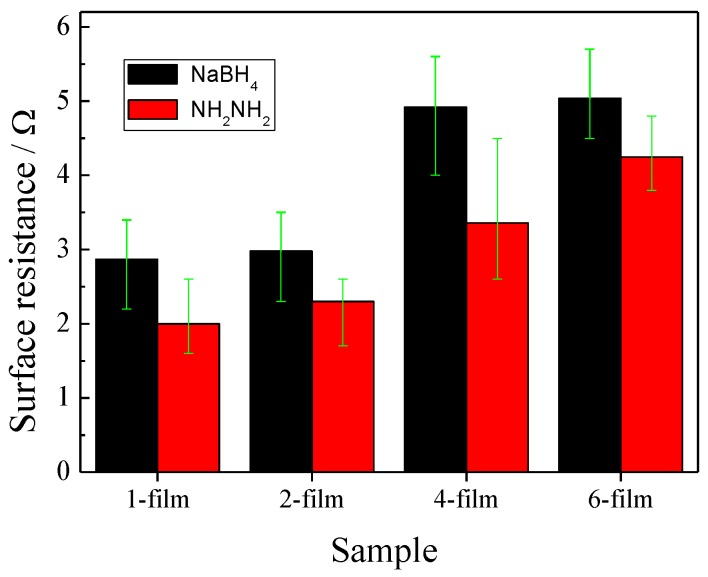
Surface resistances of the IPMCs with various thicknesses.

**Figure 5 polymers-11-00091-f005:**
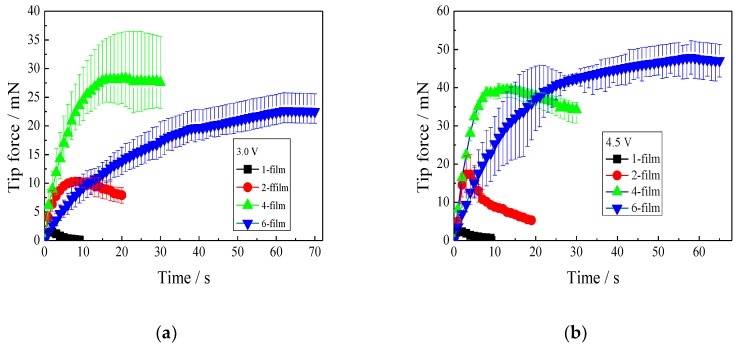
Tip forces of the IPMCs with various thicknesses under different voltages. (**a**) 3.0 V; (**b**) 4.5 V; (**c**) relationships between the maximum tip force or the response time and the thickness.

**Figure 6 polymers-11-00091-f006:**
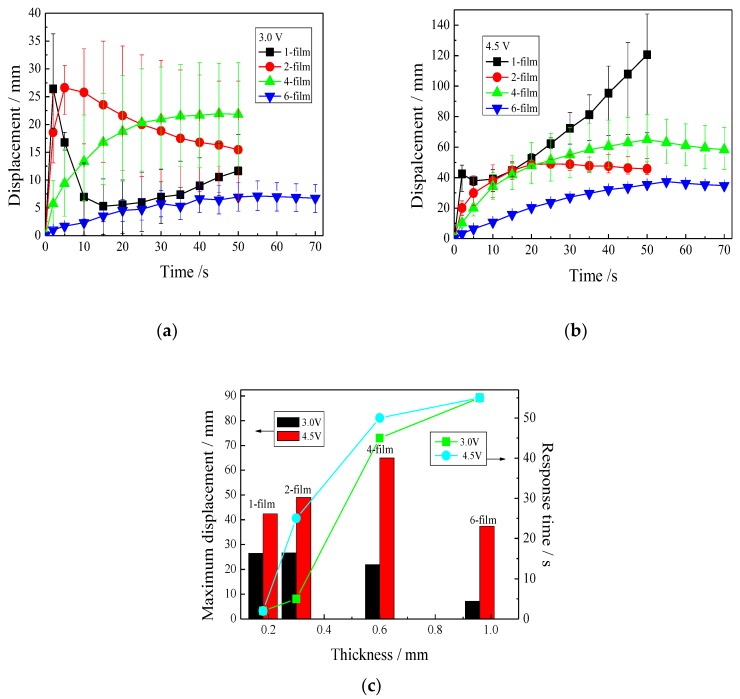
Displacements of the IPMCs with various thicknesses under different voltages. (**a**) 3.0 V; (**b**) 4.5 V; (**c**) relationships between the maximum displacements or the response time and the thicknesses.

**Figure 7 polymers-11-00091-f007:**
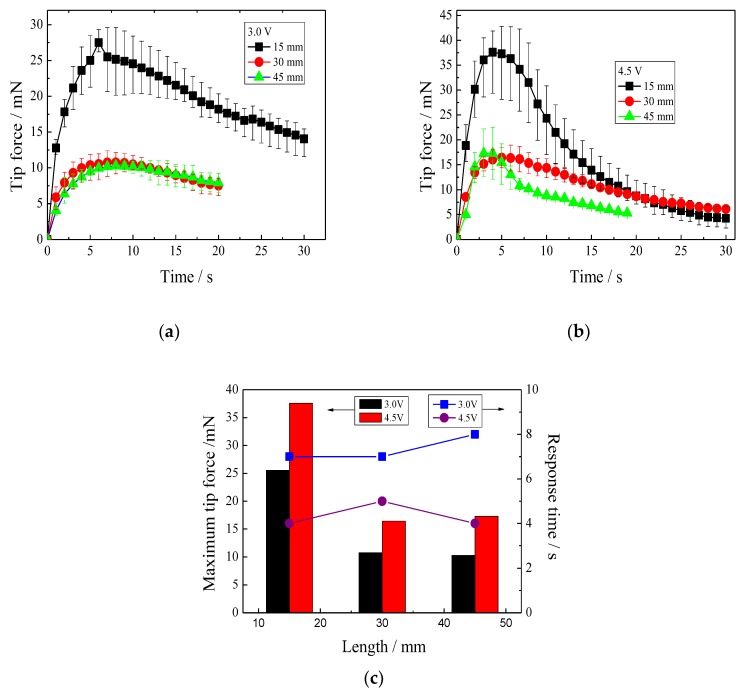
Tip forces of the IPMCs, with various lengths under different voltages. (**a**) 3.0 V; (**b**) 4.5 V; (**c**) relationships between the maximum tip forces, or the response times and lengths.

**Figure 8 polymers-11-00091-f008:**
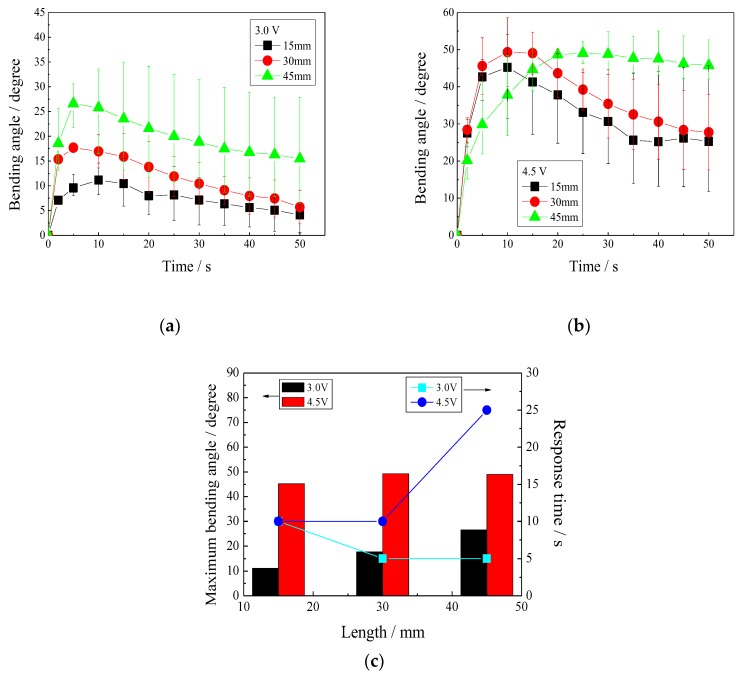
Bending angles of the IPMCs with various lengths under different voltages. (**a**) 3.0 V; (**b**) 4.5 V; (**c**) relationships between the maximum bending angles or the response time and the lengths.

**Figure 9 polymers-11-00091-f009:**
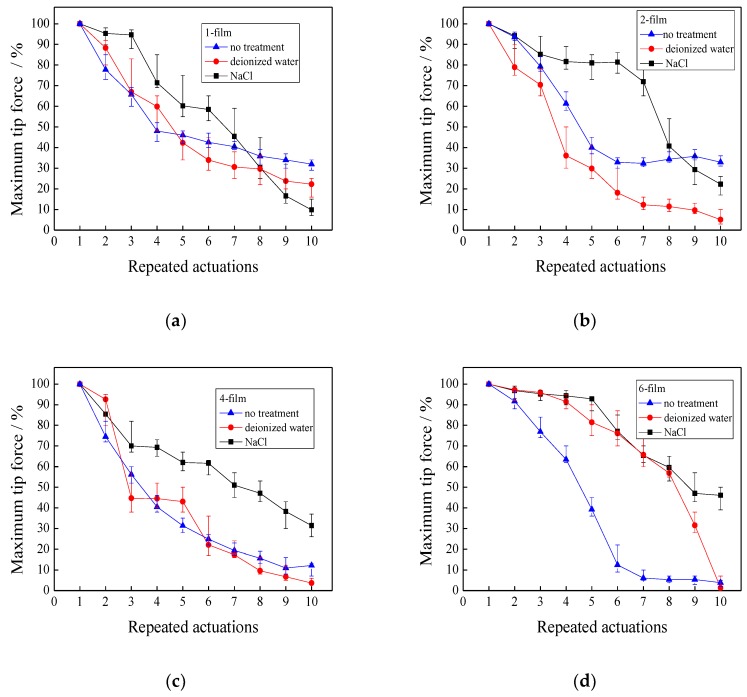
Maximum tip forces when the IPMCs with various thicknesses are repeatedly actuated. (**a**) 1-film; (**b**) 2-film; (**c**) 4-film; (**d**) 6-film.

**Figure 10 polymers-11-00091-f010:**
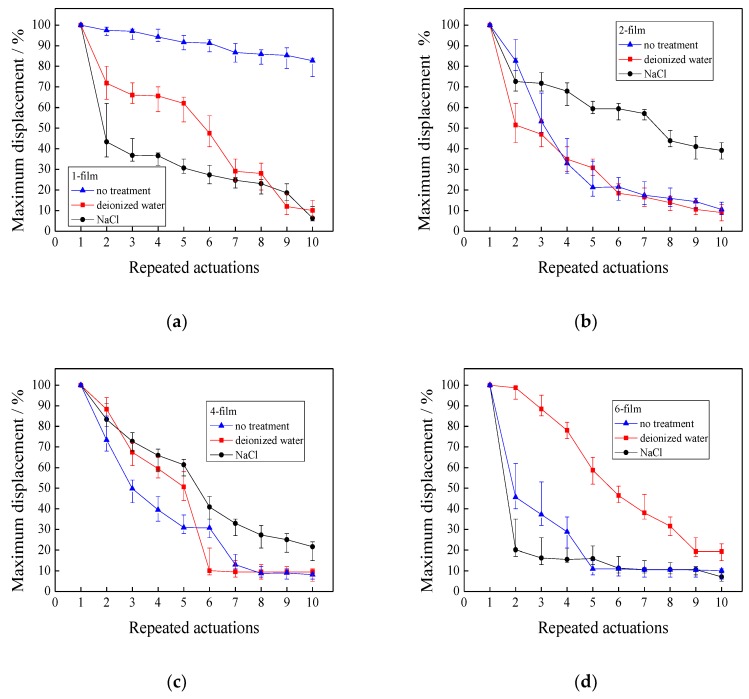
Maximum displacements when the IPMCs with various thicknesses are repeatedly actuated. (**a**) 1-film; (**b**) 2-film; (**c**) 4-film; (**d**) 6-film.

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
