# Peer review of "Characterization and Actuation of Ionic Polymer Metal Composites with Various Thicknesses and Lengths"

_polymers, 2019, doi:10.3390/polym11010091_

Round 1

Reviewer 1 Report

In this manuscript, the authors report fabrication of IPMC actuators with different thicknesses by hot pressing multiple layers of Nafion together. They varied multiple parameters and measured their effect on actuator displacement and tip force. IPMC actuators are a promising class of soft actuators and this is a timely study. However, the paper mainly lists experimental results with not much analysis. Some of the observed trends and effects might be easily explained by existing theory. Others might require deeper analysis outside of the scope of this paper. I can recommend the paper for publication once some more analysis has been added. In particular, I have the following comments:

6-film actuators are observed to have the lowest Pt content. Do the authors have any explanation for this?

The exponential fit in figure 3 only makes sense if there is an explanation. Otherwise, the line is just a guide for the eye.

The resistance in figure 4 should be quoted as sheet resistance.

In figures 5 and 6, the 6-film sample doesn’t reach its peak during the experimental time frame. The authors should extend the measurement time until they see a peak and update part c of the figures and its discussion.

The bar graphs in part c of many figures has two bars for the different voltages stacked on top of each other. This makes it difficult to compare values for 4.5V. The two bars should be placed next to each other.

Effects of thickness and length on displacement and tip force should be discussed in terms of beam theory to understand experimental results and separate the effects of geometry and material. The following papers have reported similar modeling for IPMC actuators and might be useful:

Lee, S., Park, H. C. & Kim, K. J. Equivalent modeling for ionic polymer–metal composite actuators based on beam theories. Smart Materials and Structures 14, 1363–1368 (2005).

Grau, G., Frazier, E. J. & Subramanian, V. Printed unmanned aerial vehicles using paper-based electroactive polymer actuators and organic ion gel transistors. Microsystems & Nanoengineering 2, 16032 (2016).

What is the effect of voltage on stability and reusability?

Reviewer 2 Report

This paper investigated the characterization and actuating performs of 1, 2, 4 and 6-films IPMC actuators. The following points should be changed before publication.

1.        Tip force: Units should be SI units. Please change gf to mN.

2.        The term of “Angle displacement” is strange. Angle is not displacement. “Root angle” is better.

3.        In Figure 10, the vertical axes seems to be not loss values. The title of vertical axes should be changed.

Round 2

Reviewer 1 Report

The authors have addressed most of my comments. The level of analysis to explain the experimental findings is still not very deep. The authors justify this with the maximum length of the paper and plan to publish a follow-on paper. I would prefer this paper to be longer with more analysis but if this violates the length requirements of the journal, it can be published as-is.